# Changes in Fecal Glucocorticoid Metabolites in Captive Coyotes (*Canis latrans*): Influence of Gender, Time, and Reproductive Status

**DOI:** 10.3390/ani13233596

**Published:** 2023-11-21

**Authors:** Eric M. Gese, Patricia A. Terletzky, Cole A. Bleke, Erika T. Stevenson, Susannah S. French

**Affiliations:** 1U.S. Department of Agriculture, Wildlife Services, National Wildlife Research Center, Utah Field Station, Logan, UT 84322-5230, USA; etstevenson85@gmail.com; 2Department of Wildland Resources, Utah State University, Logan, UT 84322-5230, USA; pat.terletzky@usu.edu (P.A.T.); bleke.cole@gmail.com (C.A.B.); 3Department of Biology, Utah State University, Logan, UT 84322-5305, USA; susannah.french@usu.edu

**Keywords:** breeding, canid, *Canis latrans*, carnivore, cortisol, coyote, fecal glucocorticoid metabolites, fGCM, non-invasive

## Abstract

**Simple Summary:**

Biologists have long considered producing offspring a demanding time in the life of any animal, with reproducing and raising offspring being physiologically stressful. We examined whether breeding and producing pups was more stressful than other life-history stages among captive coyotes (*Canis latrans*) using fecal sampling and subsequent assays for glucocorticoid metabolites. Using 12 pairs of coyotes (five pairs produced pups, seven pairs did not), we examined fecal glucocorticoid metabolites (fGCM) covering 11 biological time periods for one year. We found high individual variability among both females and males with no apparent statistical effect of reproduction on fGCM concentrations across the time periods; however, fGCMs spiked in mid-gestation for both pregnant and nonpregnant females, indicating some level of stress of producing pups. Levels of fGCM’s were highest among females regardless of reproductive status as compared to male coyotes. Knowledge of factors influencing fGCM concentrations among captive animals can assist in the interpretation of levels found in free-ranging animals.

**Abstract:**

Reproduction is considered an energetically and physiologically demanding time in the life of an animal. Changes in physiological stress are partly reflected in changes in glucocorticoid metabolites and can be measured from fecal samples. We examined levels of fecal glucocorticoid metabolites (fGCMs) in 24 captive coyotes (*Canis latrans*) to investigate responses to the demands of reproduction. Using 12 pairs of coyotes (five pairs produced pups, seven pairs did not), we analyzed 633 fecal samples covering 11 biological periods (e.g., breeding, gestation, and lactation). Levels of fGCMs showed high individual variability, with females having higher fGCM levels than males. The production of pups showed no statistical effect on fGCM levels among females or males. Among females, fGCM levels were highest during 4–6 weeks of gestation compared to other periods but were not significantly different between pregnant and nonpregnant females. Among males, the highest fGCM levels were during 1–3 weeks of gestation compared to other periods, but were not significantly different between males with a pregnant mate versus nonpregnant mate. Of females producing pups, litter size did not influence fGCM levels. Given that they were fed ample food throughout the year, we found that the demands of producing pups did not appear to statistically influence measures of fGCM concentrations in captive coyotes.

## 1. Introduction

Glucocorticoids (GC) and their glucocorticoid metabolites (GCMs), including cortisol and corticosterone, are hormones involved in regulating energy [1] and assist most mammals [2], including carnivores [3,4,5,6,7], to overcome stressful or metabolic changing situations [2,8,9]. Stress is a cascade of neurological, hormonal, and immunological responses to changes in the environment [10] and includes daily responses to metabolic situations such as digestion and sleeping, as well as long-term challenges such as reproduction [9]. Multiple factors (e.g., age, breeding status, environmental conditions, food quality and quantity, social status, and species) influence GCM levels [1], and there is no general GCM profile for how wild animals respond to stress [11].

Acute stressors may provide beneficial effects on individuals by promoting behaviors and physiology that help the individual cope with unexpected challenges [1] and by facilitating appropriate responses to stressors (e.g., flight from a predator). During acute stress, GCs and GCMs facilitate physiological changes, allowing individuals to rapidly respond to the stressor through increased heart rate, energy, and blood flow to muscles [1]. Conversely, more long-term, or “chronic stress”, can lead to detrimental effects, including inhibition of normal reproductive function, immune system suppression, and tissue atrophy [1,12]. Although multiple factors can influence GC and GCM levels, individual variation influences stress response more than other factors [1,11,13,14] and makes identification of a definitive pattern of stress responses challenging. Regardless of why GC or GCM levels are high, moderate, or low, the ultimate objective for an individual is to increase survival and reproduction (i.e., fitness). Rather than a linear relationship between GC and survival, researchers [1] proposed a log-quadratic relationship where intermediate levels of GCs are associated with higher survival [8]. 

Adrenocorticotropic hormone challenges in carnivores, including African wild dogs (*Lycaon pictus*), gray wolves (*Canis lupus*), and coyotes (*Canis latrans*), have provided information about physiological stress in predators [4,15,16] and suggested GCMs influence carnivores in similar ways as other mammals. The influence of stress on GC and GCM levels in carnivores is complex, with some studies suggesting that chronic stress does not negatively influence reproduction in carnivores [17,18], while others have found that GC levels were negatively influenced by group composition, land use by humans, proximity to people, and even impacts from prolonged drought [19,20,21]. Additionally, measures of glucocorticoid metabolites may not in themselves always be good indicators of stress, hence explaining the contradictory results found from different studies. 

Both females and males excrete more GCMs during reproductive seasons than non-reproductive seasons, with differences in GCM levels between females and males occurring in some species. For example, female red foxes (*Vulpes vulpes*) had greater variation in peak GCMs than male foxes [22], while there were no differences in fecal GCM levels between female and male bat-eared foxes (*Otocyon megalotis*; [23]). Conversely, an individual’s sex and age influenced GCM levels in Mexican grey wolves (*Canus lupus baileyi*; [24]). 

Examination of GC and GCM levels at a broad temporal scale indicated that there were changes between the breeding and non-breeding seasons [1,24,25,26,27], with most mammalian species exhibiting an increase in GCs and GCMs during the reproductive season, late pregnancy [28], and lactation. While polar bears (*Ursus arctus*) in zoos did not exhibit seasonal differences [29], domestic cats (*Felis catus*; [30]), Amur wildcats (*Prionailurus bengalensis euptilura*; [31]), and Eurasian lynx (*Lynx lynx*; [32]) demonstrated a consistent increase in hair cortisol values during the mating season. The caracal (*Caracal caracal*), which is not a seasonal breeder (i.e., can breed any time of the year), exhibited a higher increase in hair cortisol during the autumn and lower during the spring [31]. In red wolves (*Canis rufus*), there was no sex difference in GCM levels for females and males producing pups and those not producing pups [20].

Measures of physiological stress may be obtained from invasive sampling (e.g., blood and tissue) or non-invasive sampling (e.g., hair and feces). The process of handling animals to obtain biological samples results in the release of GCs and a concomitant spike in stress hormones, resulting in a biased measure of physiological stress [8]. Collecting carnivore scats is a non-invasive means of measuring GCM levels without the associated increase in stress hormones caused by handling [8,33,34]. Collecting scats and the resulting analysis of fecal GCMs (fGCMs) have been conducted since the early 2000s and results in little to no physiological feedback [8,35]. Several studies have examined levels of fGCMs in carnivores [7], including coyotes [4,36], bat-eared foxes [23], and Mexican gray wolves [24]. Non-invasive fecal sampling was used to explore the causes of low reproduction in female red wolves [20].

Since multiple factors may influence measures of stress hormones, we designed the current study to increase our understanding of the factors influencing physiological stress in captive coyotes. Using non-invasive sampling, the goal of our study was to determine how fGCM levels in captive coyote pairs, with or without pups, responded to seasonal changes in biological demands (e.g., breeding, gestation, and lactation). The objectives of this study were to (1) investigate temporal changes in fGCM levels across biological time periods throughout the year, (2) assess the influence of breeding status and gender on fGCM levels, and (3) examine if litter size affected fGCM levels in captive female coyotes. We hypothesized that fGCM concentrations would vary by time and breeding status, that females would have higher concentrations than males during most life-history stages, and that larger litter sizes would increase fGCMs in females. We predicted that females producing pups would also have higher fGCM concentrations than females without pups, particularly during reproductive life-history stages.

## 2. Materials and Methods

### 2.1. Study Subjects and Sample Collection

We housed 24 coyotes in 0.10 ha outdoor pens at the U.S. Department of Agriculture National Wildlife Research Center’s Predator Research Facility, Millville, Utah; see [4] for further housing details. We moved the coyote pairs into their pens at least three days prior to scat collections, and pens were cleared of scats the day prior to collections. The coyotes were fed a commercially prepared carnivore diet (Fur Breeders Agricultural Cooperative, Sandy, Utah) once daily and fasted one day per week. Water was provided ad libitum. We added glitter (Glitterex Corporation, Cranford, NJ, USA) to their food, differentiated by gender, to ensure knowledge of which individual a scat sample was collected from. We collected fresh fecal samples each week for 12 months from the 24 captive coyotes (2 to 7 years old) divided into five pairs that produced pups and seven pairs that did not produce pups. Six pairs not producing pups contained males that were vasectomized > 1 year prior to the study; the other pair had an intact male but failed to produce pups. Breeding pairs were fed extra food during gestation and lactation to compensate for the increased energy costs of pup production [37]. We collected fecal samples from coyote pairs randomly each time to reduce biasing physiological measures. Samples were frozen following collection at −80 °C.

### 2.2. Time Periods for Analysis

We separated the 12-month collection period into 11 biological time periods coinciding with life-history stages (i.e., breeding, gestation, lactation, weaning, pup removal, and non-reproductive periods [diestrus]). Time periods were back and forward calculated based on the known whelping date (Table 1). Pair bonding season occurred from 1 November to 31 December, and breeding season occurred from 1 January to the day before conception. Coyote gestation is around 63 days [38,39]; thus, the conception date for each breeding pair was 63 days prior to the known whelping date. The next period was gestation, which began one day following conception. We divided gestation into three trimesters (i.e., first [1–3 weeks], second [4–6 weeks], and third [7–9 weeks]) to assess fine-scaled temporal fGCM concentrations. Lactation spanned six weeks following whelping and was divided into two 3-week periods (i.e., early [1–3 weeks], later [4–6 weeks]). Weaning occurred one day after lactation ended and extended until the pups were removed from the pens [40], which occurred at 8–10 weeks of age. The period following pup removal was divided into two 3-week periods (i.e., early [1–3 weeks], later [4–6 weeks]). The non-reproductive period (diestrus) started 7 weeks after pup removal on 31 October. Pairs without pups were divided into time periods based on the average dates of the reproductive pairs.

### 2.3. Laboratory Methodologies

We extracted steroid metabolites from the fecal samples using a phosphate and methanol wet extraction buffer [41]. We weighed 20 mL scintillation vials (Wheaton, Millville, NJ, USA) prior to use, with the lids off to constitute the initial weight. Each fecal sample was thawed, weighed to 0.50 ± 0.01 g wet feces, added to a labeled vial, and inundated with 5.0 mL of working fecal extraction buffer (50:50 buffer:methanol). Using a clean spatula, we broke up the fecal to incorporate the buffer and vortexed for 20 s. Vials were placed on a shaker at 200 rpm for a minimum of 16 h. The liquid was decanted into 12 × 75 mm glass tubes (Fisherbrand, Fisher Scientific, Ottawa, ON, Canada) after the vials were allowed to settle for an hour. Following the decanting, the tubes were centrifuged for an hour at 3500 rpm at 4 °C (Beckman Coulter Allegra™ 6R Centrifuge, Indianapolis, IN, USA). Tubes were decanted again into clean 1.5 mL microcentrifuge tubes (Fisherbrand) and stored at −80 °C. The remaining fecal in the glass tubes was poured back into the original scintillation vials and placed in a drying oven until all moisture was evaporated. Vials were weighed a second time, constituting the sample’s final weight to standardize GCM concentrations based on fecal weight per sample. 

We measured steroid hormones using enzyme-linked immunosorbent assay kits (ELISA, Enzo Life Sciences, Inc., Farmingdale, New York, NY, USA) using the manufacturer’s protocol for cortisol and optimized for coyotes, followed by assay methodologies for species validation [42]. The cortisol ELISA is based on competitive binding between mouse monoclonal antibodies and plasma hormone that occurs on a goat anti-mouse immunoglobulin microtiter plate. We labeled 12 × 75 mm glass tubes to generate standard curves. The volume of assay buffer and hormone standard was 1:20. The minimum detectable value for cortisol was 0.05672 ng/mL. We adjusted the dilution of the sample for individuals whose output failed to fall on the standard curve. A standard curve was run for each assay. Assay kits were run in duplicate with 37 per plate, which were averaged to calculate the resulting concentration for each sample. We pipetted standard dilutions, controls, and zeros (blanks to control for non-specific binding) onto the plate and added assay buffer. Each plate was incubated at room temperature on a titer plate shaker (Lab-Line Instruments, Inc., Kerala, India) for two hours at 500 rpm. After shaking, we washed each plate four times with the supplied wash buffer in a plate washer (BioRad, Hercules, CA, USA). Next, we added p-nitrophenyl phosphate (pNpp) substrate solution to every well and incubated it with shaking for 45–60 min at room temperature. Finally, we added a stop solution to every well and read the plates immediately using a microplate reader (BioRad) with optical density and correction. Standard curves, controls, and controls for non-specific binding were run on each plate. We displayed standard curves as four-parameter logistic curves.

Immunoassays for measuring fecal hormone metabolite levels need to be validated in a species-specific manner to ensure that the hormone of interest is being properly measured as a metabolite in the feces [8]. Due to the novelty of assay use in this species, we completed fecal hormone validations for coyotes prior to running samples. Validations were established to determine appropriate hormone concentrations, control for assay precision, and detect any potential non-specific binding that would bias results. We validated fecal glucocorticoid metabolites in female and male coyotes. First, we determined parallelism for the hormone validation. Second, we used an extracted sample from the same individual and generated a spectrum of high to low volumes to associate with the standard curve. We began with using 100 μL of extraction solution and diluting in half until 6.25 μL. Further dilution was necessary if samples were not on the standard curve. All assay wells during this process were of the same individual. Third, we added spikes to calculate recovery to ensure only the hormone of interest was being measured and no binding interference was occurring. The final step was calculating intra- and inter-assay variation, which serves as a level of certainty that generated values can be compared with each other. Cortisol validation and measurements were conducted across 18 assays with an intra-assay variation of 4.9% and an inter-assay variation of 35.0%. Assay spikes and recovery tests (recovery = 108%), in addition to parallelism curves (R^2^ = 0.99), were completed for species validations of fGCMs.

### 2.4. Data Analysis

Based on information in the literature, the distinct differences in fGCM levels between females and males (Figure 1), and apparent biological differences (mainly females can get pregnant, males cannot), we conducted separate analyses on females and males. We conducted linear regressions [43] of litter size against female fGCM levels during the three trimesters of gestation and early and late lactation for females with pups. We compared fGCM concentrations of females and males with pups and without pups for each time period to assess if there was a reproductive effect. If the time period dataset met assumptions of normality and constant variances, we used a *t*-test [43] to test for differences between fGCM levels of females and males with and without pups. If assumptions of normality and constant variance were not met, we used a Wilcoxon signed-ranked test [43] to assess differences in fGCM levels of coyotes with and without pups. To determine if there were time period differences, we conducted a repeated measures analysis of variance (ANOVA; [43]) with fGCM levels as the ‘response’ variable, individuals as the ‘blocking’ variable, and time periods as the ‘within subject’ variable. The repeated measures ANOVA was conducted in R [44] with the ggplot2 [45], ggpubr [46], tidyverse [47], and rstatix [48] packages. If there was a time period effect on fGCM levels, a Tukey’s Honest Significant Difference (HSD; [43]) was conducted to identify which time periods were significantly different. We determined statistical significance when *p* < 0.05.

## 3. Results

We assayed 633 scats from 12 pairs of captive coyotes in two groups (pups produced versus no pups produced) covering 12 months. Of the six intact males, one male (M0423) did not produce pups with the paired female (F0580); thus, we analyzed five pairs with pups and seven pairs without pups. When comparing the genders, individual female fGCM values were consistently higher, exhibited more individual variation, and had greater variation in individual maximums than males (Figure 1 and Figure 2). The highest fGCM concentration (28,610 ng/g) was in a female without pups and occurred during 4–6 weeks of gestation, while the second highest concentration (28,528 ng/g) was in a female with pups and occurred during the first 1–3 weeks of gestation (Figure 3). The highest fGCM concentration for a male (13,825 ng/g) occurred during the first 1–3 weeks of gestation for a male with pups, as was the second highest fGCM concentration (7180 ng/g) for a male with pups and occurred during 4–6 weeks of gestation (Figure 4). Both genders exhibited high individual variability. High variability among individuals was pronounced among both females (Figure 3) and males (Figure 4), with no clear trend that the variability was related to the location of the pen (i.e., fGCMs levels were not correlated between mates sharing the same pen).

Within females, those with pups had lower average fGCM levels (2322 ± 3170 ng/g) than females without pups (3030 ± 4637 ng/g), but there was no difference in fGCM concentrations for females with pups versus those without pups in any time period (Table 2). We did find there was a time period effect when individuals were considered in a repeated measures ANOVA (*F* = 4.86, *p* < 0.01), with Tukey’s post hoc analysis showing female fGCMs values during mid-gestation (4–6 weeks) were significantly higher (*p* < 0.01) than all other periods (Table 3); the repeated measures indicated no ‘treatment effect’ of pups versus no pups, but did show a ‘time period’ effect. The average litter size for the pairs producing pups was 3.4 pups (range = 2–6 pups), and there were no significant relationships (all *p*-values > 0.13) between fGCM levels in females and litter size during the time periods of gestation or lactation.

Within the males, those with pups had an average fGCM of 1507 ng/g, while males without pups had an average fGCM concentration of 1051 ng/g, but there was no difference in fGCMs for males with pups versus without pups in any time period (Table 2). In contrast to females, the males had relatively constant variability and maximum fGCM values amongst all individuals (Figure 1). There was a time period effect when individuals were considered in the repeated measures ANOVA (*F* = 3.64, *p* < 0.01), with Tukey’s post hoc analysis showing there were significant differences (*p* < 0.03) in fGCM values between early gestation (1–3 weeks) and all other time periods, except late gestation (7–9 weeks; Table 4). The repeated measures indicated no ‘treatment effect’ of pups versus no pups but simply a ‘time period’ effect. Levels of fGCMs during early gestation exhibited the highest amount of individual variation among the males (Figure 2).

## 4. Discussion 

We found no statistically significant effect of the demands of producing pups on fGCM concentrations in female and male captive coyotes. However, while not statistically different, levels of fGCM concentrations did spike during mid-gestation and lactation among females who were pregnant and not pregnant (Figure 3). However, the high variability among individual animals made detecting statistical significance difficult. Among the reproducing females, we found no relationship between fGCM levels and litter size. However, our coyotes did not experience any food limitations, which would likely reduce the possibility of an influence of pup production on fGCM concentrations. Among two carnivores, studies revealed no relationship between cortisol levels and litter size. For example, female fisher (*Pekania pennanti*; [21], which can breed once a year, and domestic cats [30], which can breed several times a year, did not exhibit any relationship between cortisol levels measured in hair and litter size. In contrast, captive cheetah females with a single cub had higher fGCMs than those with multiple cubs [9], suggesting that higher metabolic costs and cortisol levels before or during breeding reduced the amount of energy available to developing oocytes, and thus, fewer embryos were released. Wolves and bears (*Ursus* spp.) in the wild experienced an increase in cortisol levels when prey density and food availability were low [26,27].

Similar to other species, we found different fGCM levels between females and males, with females exhibiting greater concentrations and higher variability than males [24,49,50,51]; although one study found little variability between sexes [23]. Trends across sexes are generally not consistent among carnivores and are likely influenced by food quantity and quality, pregnancy status, and whether the individual is actively breeding. For example, female tigers (*Panthera tigris*) in India in lower-quality habitats with a low reproductive rate had higher fGCM values than those in higher-quality habitats [52]. During reproductive seasons, GCMs can vary greatly among sexes, age classes, and individuals, but there is some suggestion that participating in raising young can be less stressful, as indicated by individuals who participated in alloparenting [51]. Male lynx (*Lynx canadensis*) exhibited an increase in fGCM levels at the start of the reproductive season, while fGCM levels of females increased during the initial stages of pregnancy [50]. While male dingoes (*Canis dingo*) with pups had higher levels of GC than those without pups [53], there was no such trend for females [51]. Pair-bonding can also be a stressful time for receptive females, as males continually harass them to copulate, and the stress can result in increased GCM levels, as indicated in captive red wolves [20]. In terms of fitness consequences, female fishers in California with low cortisol levels experienced increased survival compared to females with medium and high cortisol levels, but there was no relationship between survival rates of males and cortisol concentrations [21].

The high variability among individual females and males likely influenced our ability to find statistical differences in fGCM levels across the time periods between pairs with pups versus those without pups. However, there was a time effect, with females having the highest levels of fGCM concentrations during gestation and again in lactation, suggesting some level of stress while pregnant and nursing the pups. Interestingly, these concentrations were the same between pregnant and nonpregnant females. Nonpregnant female coyotes do undergo a pseudopregnancy with no differences in mating behaviors and progesterone levels between pregnant and nonpregnant females, with similarly high individual variability in reproductive hormone levels before and during breeding [54]. Whether pheromones play a role in nonpregnant coyotes exhibiting similar changes in fGCM levels as the pregnant coyotes deserve further investigation. The pairs were placed next to both pairs having pups and pairs not having pups to eliminate potential biases of where they were housed in pens separated by <5 m and had 2 m high cement walls between each pen, which would allow pheromones to infiltrate adjacent pens.

While variability among individuals can reduce distinct patterns of fGCMs, predators, in general, show higher fGCMs during the breeding season than the non-breeding season. For example, female fishing cats (*Prionailurus viverinus*) and wolves exhibited higher fGCMs during winter (breeding season) compared to the non-breeding summer season [24,26,55], although no seasonal pattern was observed in polar bears [29] and tigers [6], suggesting that body size and/or food requirements may influence fGCM values. Seasonal GCM levels are modulated as a balance between required energy expenditure and available energy [1], but since the food available to our captive coyotes was constant, our fGCMs levels were likely not influenced by food availability or the demands of producing pups and were likely a stress response to an outside stimulus. 

While there is agreement that GC and GCM levels generally increase in response to stress, there is no consistent pattern of the response level or direction of changes among species or even within a species due to the high amount of variability among individuals [1,11]. Noise and unexpected activity can increase short-term stress [56] for captive individuals, as well as an increase in human visitations, which ultimately can increase cortisol levels. Increases in human activities can influence cortisol and GCM levels, even in individuals accustomed to human activity, such as visitor levels in zoos. Mexican wolves in zoos [57] and tigers in Indian reserves [58] exhibited higher levels of cortisol on days with higher human visitation rates. 

In our study, six of the seven males and three of the seven females without pups experienced a spike in fGCM levels on 29 May 2011, while two males and two females with pups experienced similar spikes. Thus, 13 of 24 (54%) coyotes showed a spike in fGCM concentrations from scats collected on May 29. Examining activity logs for the facility showed that the pups in all five pens in this study, plus the pups from three neighboring pens, were captured and handled as part of the routine colony duties for monitoring pup health and measuring growth rates. Many activities occur on the site related to animal care and facility maintenance; thus, individual responses to anthropogenic activities may be reflected in their fGCM levels (46% of the coyotes did not show a spike in fGCMs) and may indicate their level of tolerance or desensitization to human activities or even individual personalities (e.g., wary versus bold animals). In wild coyotes, dominant alpha coyotes exhibited high levels of wariness to remote camera sites compared to beta and transient coyotes [59]. At the same facility as this study, the behavioral responses of coyotes to light and sound stimuli were categorized as bold, shy, and persistent [60]. Similarly, an evaluation of behavioral syndromes at the same facility suggested a behavioral syndrome in coyotes for boldness and exploration [61]. Thus, individual personalities and tolerance of human activities likely explain the high variability of fGCM concentrations observed in our study.

## 5. Conclusions

This is the first and only study examining fGCM concentrations covering an entire year in the life of a coyote, tracking fGCM levels through the different life-history periods from breeding to diestrus. While we found little statistical effect of producing pups on fGCM concentrations in female and male captive coyotes, we did find spikes in fGCM levels occurred during gestation and lactation among the female coyotes, but high individual variability prevented finding statistical significance. Our results indicated that variability among individual coyotes over time was the main parameter explaining observed fGCM levels with little effect from breeding and reproduction. High variability in fGCM levels among the coyotes likely reflected individual personalities and tolerance of human activities occurring on-site or nearby. Since there are multiple external (e.g., environment, food quality, and prey density) and internal factors (e.g., age, breeding status, social rank, and individual personality) that can influence GCM levels in a particular species [1,11,13,14], understanding the factors influencing changes and variability of fGCM levels in captive coyotes may prove useful when interpreting fGCM concentrations measured in captive coyotes in different artificial environments as well as potential application to free-ranging animals. We recognize the limitations of using captive animals to infer fGCM measures from wild animals, especially when wild coyotes undergo bottlenecks in food availability, particularly during the winter when females become pregnant, and food limitations can severely affect litter size [62,63,64,65,66].

## Figures and Tables

**Figure 1 animals-13-03596-f001:**
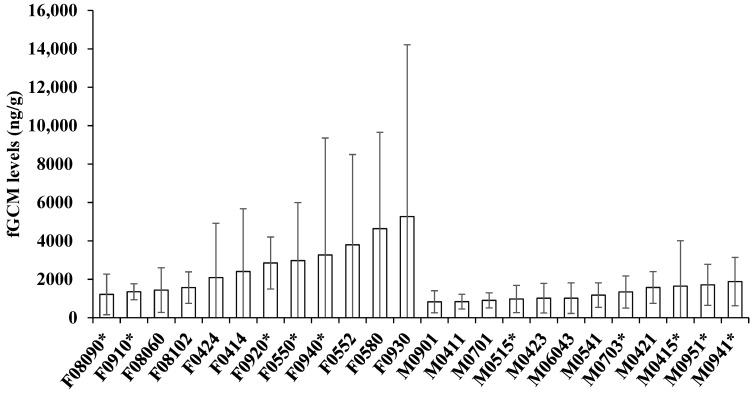
Mean and standard deviation of fecal glucocorticoid metabolite (fGCM) levels for captive females (F) and males (M); * indicates individuals with pups.

**Figure 2 animals-13-03596-f002:**
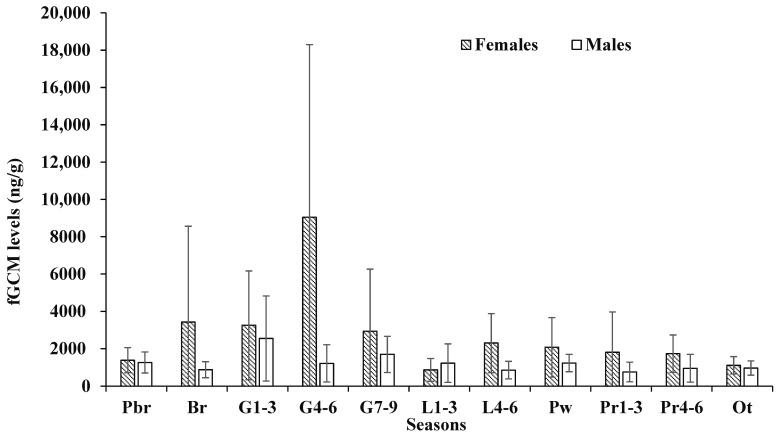
Mean and standard deviation of seasonal fecal glucocorticoid metabolite (fGCM) levels for captive female and male coyotes across 12 months. Time periods are pair bonding and reinforcement (Pbr), mate attraction and copulation (Br), 1–3 weeks into gestation (G1–3), 4–6 weeks into gestation (G4–6), 7–9 weeks into gestation (G7–9), 1–3 weeks into lactation (L1–3), 4–6 weeks into lactation (L4–6), after pups are weaned (Pw), 1–3 weeks after pups are removed (Pr1–3), 4–6 weeks after pups are removed (Pr4–6), and the non-reproductive season (Ot).

**Figure 3 animals-13-03596-f003:**
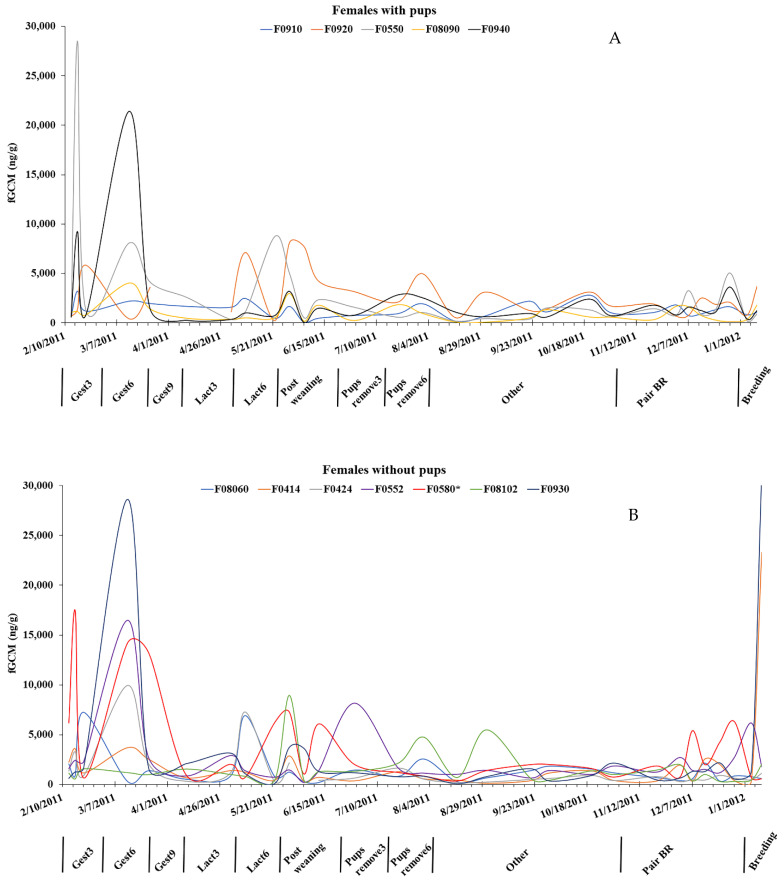
Fecal glucocorticoid metabolite (fGCM) levels for female coyotes (**A**) with pups and (**B**) without pups across 12 months. Time periods are: 1–3 weeks into gestation (Gest3), 4–6 weeks into gestation (Gest6), 7–9 weeks into gestation (Gest9), 1–3 weeks into lactation (Lact3), 4–6 weeks into lactation (Lact6), after pups are weaned (Post weaning), 1–3 weeks after pups are removed (Pups remove3), 4–6 weeks after pups are removed (Pups remove6), non-reproductive season (Other), Pair bonding and reinforcement (Pair BR), and mate attraction and copulation (Breeding). * indicates animal was originally in the breeding cohort but failed to produce pups, thus classed as a female without pups.

**Figure 4 animals-13-03596-f004:**
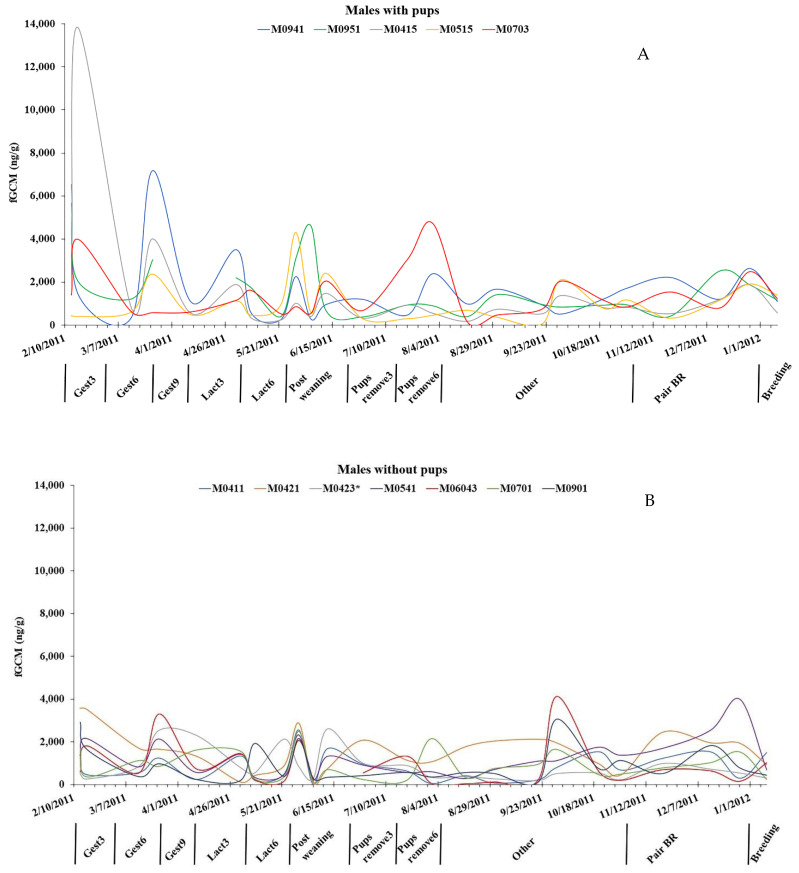
Fecal glucocorticoid metabolite (fGCM) levels for male coyotes (**A**) with pups and (**B**) without pups across 12 months. Time periods are 1–3 weeks into gestation (Gest3), 4–6 weeks into gestation (Gest6), 7–9 weeks into gestation (Gest9), 1–3 weeks into lactation (Lact3), 4–6 weeks into lactation (Lact6), after pups are weaned (Post weaning), 1–3 weeks after pups are removed (Pups remove3), 4–6 weeks after pups are removed (Pups remove6), non-reproductive season (Other), Pair bonding and reinforcement (Pair BR), and mate attraction and copulation (Breeding). * indicates animal was originally placed in the breeding cohort but failed to produce pups, thus was classed as a male without pups.

**Table 1 animals-13-03596-t001:** Dates of each time period for each pair calculated from the known whelping dates. Periods are pair bonding and reinforcement (Pbr), mate attraction and copulation (Br), 1–3 weeks into gestation (Gest1–3), 4–6 weeks into gestation (Gest4–6), 7–9 weeks into gestation (Gest7–9), 1–3 weeks into lactation (Lact1–3), 4–6 weeks into lactation (Lact4–6), after pups are weaned (Pw), 1–3 weeks after pups are removed (Pr1–3), 4–6 weeks after pups are removed (Pr4–6), and the non-reproductive season (Other).

Period	With pups					Without pups
Event	F0910/M0941	F0920/M0951	F0550/M0415	F08090/M0515	F0940/M0703	
Pbr	11/1–12/31	11/1–12/31	11/1–12/31	11/1–12/31	11/1–12/31	11/1–12/31
Br	1/1–2/13	1/1–2/6	1/1–2/10	1/1–1/31	1/1–2/1	1/1–2/6
Fertilization	2/13	2/6	2/10	1/31	2/1	2/6
Gest1–3	2/14–3/7	2/7–2/28	2/11–3/4	2/1–2/22	2/2–2/23	2/7–2/28
Gest4–6	3/8–3/28	2/29–3/21	3/5–3/25	2/23–3/15	2/24–3/16	2/29–3/21
Gest7–9	3/29–4/17	3/22–4/10	3/26–4/14	3/16–4/4	3/17–4/5	3/22–4/10
Whelping	4/17	4/10	4/14	4/4	4/5	4/10
Lact1–3	4/17–5/8	4/10–5/1	4/14–5/5	4/4–4/25	4/5–4/26	4/10–5/1
Lact4–6	5/9–5/29	5/2–5/22	5/6–5/26	4/26–5/16	4/27–5/17	5/2–5/22
Pw	5/30–6/23	5/23–6/22	5/27–6/27	5/17–6/18	5/18–6/18	5/23–6/22
Pups removed	6/24	6/23	6/28	6/19	6/19	6/23
Pr1–3	6/24–7/15	6/23–7/14	6/28–7/19	6/19–7/10	6/19–7/10	6/23–7/14
Pr4–6	7/16–8/5	7/15–8/4	7/20–8/9	7/11–7/31	7/11–7/31	7/15–8/4
Other	8/6–10/31	8/5–10/31	8/10–10/31	8/1–10/31	8/1–10/31	8/5–10/31

**Table 2 animals-13-03596-t002:** Differences between fecal glucocorticoid metabolite (fGCM) levels (ng/g) for females and males with pups and without pups. If assumptions of normality of data and equal variances were met, a parametric *t*-test (*t*) was performed; otherwise, a Wilcoxon rank test (*W*) was performed.

	Females				Males			
Season	No Pups	Pups	*t/W*	*p*	No Pups	Pups	*t/W*	*p*
Pre-breeding	1367 ± 825	1425 ± 446	*t* = −0.14	0.89	1162 ± 682	1411 ± 336	*t* = −0.75	0.47
Breeding	5057 ± 6375	1141 ± 644	*W* = 22	0.53	731 ± 456	1096 ± 311	*t* = −1.54	0.15
Gestation 1–3 weeks	2859 ± 2435	3812 ± 3702	*W* = 15	0.76	1590 ± 1068	3893 ± 2945	*W* = 7	0.11
Gestation 4–6 weeks	10,632 ± 10,124	6814 ± 8429	*W* = 20	0.76	857 ± 454	1725 ± 1360	*W* = 9	0.20
Gestation 7–9 weeks	3507 ± 4330	2139 ± 960	*W* = 17	0.99	1819 ± 899	1523 ± 11,208	*t* = 0.51	0.62
Lactation 1–3 weeks	990 ± 621	668 ± 617	*W* = 20	0.31	1006 ± 774	1625 ± 1409	*W* = 10	0.53
Lactation 4–6 weeks	2275 ± 1291	2345 ± 2086	*t* = −0.07	0.94	777 ± 250	975 ± 698	*t* = −0.70	0.50
Post-weaning	1959 ± 1602	2244 ± 1740	*W* = 15	0.75	1063 ± 240	1481 ± 607	*W* = 10	0.27
Pup removal 1–3 weeks	2160 ± 2706	1314 ± 1162	*W* = 20	0.76	884 ± 607	583 ± 378	*W* = 23	0.43
Pup removal 4–6 weeks	1479 ± 924	2098 ± 1088	*W* = 10	0.27	715 ± 318	1290 ± 1059	*W* = 11	0.34
Non-reproductive	1048 ± 442	1212 ± 523	*t* = −0.59	0.57	950 ± 438	1002 ± 333	*t* = −0.22	0.83

**Table 3 animals-13-03596-t003:** Matrix of results from a Tukey’s multiple comparison test following a repeated measures ANOVA for fecal glucocorticoid metabolite (fGCM) levels in female coyotes; * indicates a significant (*p* < 0.05) difference between time periods.

	Pre-breeding	Breeding	Gestation 1–3	Gestation 4–6	Gestation 7–9	Lactation 1–3	Lactation 4–6	Post weaning	Pup remove 1–3	Pup remove 4–6	Other
Pre-breeding											
Breeding											
Gestation 1–3											
Gestation 4–6	*	*	*								
Gestation 7–9				*							
Lactation 1–3				*							
Lactation 4–6				*							
Post weaning				*							
Pup removal 1–3				*							
Pup removal 4–6				*							
Other				*							

**Table 4 animals-13-03596-t004:** Matrix of results from a Tukey’s multiple comparison test following a repeated measures ANOVA for fecal glucocorticoid metabolite (fGCM) levels in male coyotes; * indicates a significant (*p* < 0.05) difference between time periods.

	Pre-breeding	Breeding	Gestation 1–3	Gestation 4–6	Gestation 7–9	Lactation 1–3	Lactation 4–6	Post weaning	Pup remove 1–3	Pup remove 4–6	Other
Pre-breeding											
Breeding											
Gestation 1–3	*	*									
Gestation 4–6			*								
Gestation 7–9											
Lactation 1–3			*								
Lactation 4–6			*								
Post weaning											
Pup removal 1–3			*								
Pup removal 4–6			*								
Other			*								

## Data Availability

Data are archived at the U.S. Department of Agriculture’s National Wildlife Research Center and are available upon request (QA-1834).

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
