# Peer review of "Changes in Fecal Glucocorticoid Metabolites in Captive Coyotes (Canis latrans): Influence of Gender, Time, and Reproductive Status"

_animals, 2023, doi:10.3390/ani13233596_

Round 1
Reviewer 1 Report
Comments and Suggestions for Authors
While the study investigates the levels of fecal glucocorticoid metabolites (fGCMs) in captive coyotes during various reproductive stages, there are areas that could be addressed for improvement:
The study includes 24 captive coyotes, but the small sample size, especially considering only 5 pairs produced pups, limits the generalizability of the findings. A larger and more diverse sample, considering various environmental conditions, could enhance the study's robustness.
The study acknowledges high individual variability in fGCM levels but does not explore or discuss potential reasons for this variability. Providing insights into individual differences, such as age, social status, or previous reproductive history, would enrich the interpretation of results.
The methodology lacks detailed information on the techniques used for fGCM analysis. Providing specifics on the laboratory methods, validation processes, and quality controls would enhance the study's transparency and reproducibility.
The study primarily reports the results without delving deeply into the implications or potential reasons for the observed patterns. A more extensive discussion on the ecological or physiological factors influencing fGCM levels during specific reproductive periods would add depth to the interpretation.
The study briefly mentions that the coyotes were fed ample food throughout the year, but does not elaborate on potential environmental or husbandry factors that might influence stress levels. Considering factors such as enclosure size, social dynamics, or weather conditions could contribute to a more comprehensive analysis.
The study provides a snapshot of fGCM levels during specific periods but lacks a longitudinal perspective. Examining how fGCM levels change within individuals over time, especially during transitions between reproductive stages, would offer valuable insights into the dynamics of stress responses.
The captive setting might not fully replicate the conditions faced by wild coyotes. Discussing the potential implications and limitations of applying these findings to wild populations would contribute to the study's external validity.
Author Response
Reviewer #1
While the study investigates the levels of fecal glucocorticoid metabolites (fGCMs) in captive coyotes during various reproductive stages, there are areas that could be addressed for improvement:
The study includes 24 captive coyotes, but the small sample size, especially considering only 5 pairs produced pups, limits the generalizability of the findings. A larger and more diverse sample, considering various environmental conditions, could enhance the study's robustness.
REPLY: While the reviewer indicates that 24 coyotes is a small sample size to them, we point out that there has never been this large of sample examined; a previous study was on 1 coyote…Coyotes are not readily available in a captive setting and our colony is the only one in the world that provides a sample size of 24 animals providing some level of measurement of variability among individuals.
The study acknowledges high individual variability in fGCM levels but does not explore or discuss potential reasons for this variability. Providing insights into individual differences, such as age, social status, or previous reproductive history, would enrich the interpretation of results.
REPLY: We actually controlled for these variables in the choice of the animals to include in the study. They were all breeding adults with at least one previous litter of pups, and they all had the same age class (adult) and social status (breeding adult), and reproductive history.
The methodology lacks detailed information on the techniques used for fGCM analysis. Providing specifics on the laboratory methods, validation processes, and quality controls would enhance the study's transparency and reproducibility.
REPLY: We have added more details on laboratory methods for extractions, ELSA’s, and validation as requested. These additions are highlighted in yellow on lines 148-194.
The study primarily reports the results without delving deeply into the implications or potential reasons for the observed patterns. A more extensive discussion on the ecological or physiological factors influencing fGCM levels during specific reproductive periods would add depth to the interpretation.
REPLY: We have added more discussion about the potential reasons for the observed results as requested; these are highlighted in yellow throughout the discussion focusing on the highest levels observed during gestation and lactation.
The study briefly mentions that the coyotes were fed ample food throughout the year, but does not elaborate on potential environmental or husbandry factors that might influence stress levels. Considering factors such as enclosure size, social dynamics, or weather conditions could contribute to a more comprehensive analysis.
REPLY: All the animals were in the same sized pen, all exposed to the same environmental conditions, and all had the same social dynamic (mated pairs). Thus it is not possible to discuss these influences as they were held constant across all animals.
The study provides a snapshot of fGCM levels during specific periods but lacks a longitudinal perspective. Examining how fGCM levels change within individuals over time, especially during transitions between reproductive stages, would offer valuable insights into the dynamics of stress responses.
REPLY: The only spikes observed were during gestation and lactation. Again, we found no statistical significance due to the high individual variability, so not clear how we discuss transitions between stages when only two stages were different and we discuss those two (gestation, lactation).
The captive setting might not fully replicate the conditions faced by wild coyotes. Discussing the potential implications and limitations of applying these findings to wild populations would contribute to the study's external validity.
REPLY: I have added a sentence at the very end of the conclusion section addressing this comment, lines 352-355.
Reviewer 2 Report
Comments and Suggestions for Authors
Dear authors,
Thanks for submitting this paper. It is interesting information but overall need to be completed with more details and to approach your results with ponderation due to the bias and limitations of your study.
Here some comments:
Form:
- add scientific name in the title
- Always good to spell out number between 0 and 9. To change in the whole document
- Line 16: change to "...with no apparent effect of reproduction on…"
- Line 21: Change to "physiological stress"
- Line 41: metabolic
- Line 66: "are complex"
- Line 307: your study should not be considered as a reference
- Line 313: "six of the seven males…"
Content:
- Line 22: Glucocorticoid metabolism is not the only parameter influenced by stress. It is necessary to add a terms such as " ...are partly reflected..."
- Line 34: Corticoid alone is not a reliable parameters of stress therefore difficult to draw any conclusion about physiological stress
- Line 69: Gc are not alone good indicators of stress, hence explaining the sometimes contradictory results in different studies
- 2.1: This part is supposed to be about the subjects but you are explaining about subject, housing, sample collection and your timeline decision. It should be split into different parts.
- Line 117: add information about glitters (type, brand and so on) it will be interesting for the readers and for potential researcher willing to use the same method
- Line 127: add which temperature the sample where frozen at
- Line 140: it should be explained why the decision to remove the pups at 8-10 weeks of age instead of following natural dispersion between 6-9 months old for males and females staying with the parents longer.
- 2.2: Need to be more detailed: preparation of the sample pre-extraction. Steps of extraction and ELISA... Also need to explain how the method was optimised for coyotes.
- Line 189 to 191: the choice of calling the time period "early pregnancy" is confusing when associated to non pregnant animal or males in the same sentence. It would be good to call the period with a specific code in the text like you did for the graphics.
- Line 218: higher Cg during pregnancy is due to both mother and foetuses production this should be explained
- Discussion: there is no mention of the limitations of your study. It is essential that you talk about it for example small number of animals, only one sample collected per week creating gap into the data (ideally this kind of study should collect between two and three sample a week to decrease bias in variation). Study only over one breeding season... Gc not the ideal parameters for stress evaluation and mentioning about the importance of holistic approach via the use of the allostatic load instead.
-Line 266: you mention wild cheetah when the paper you reference to is about captive animals.
- Line 292: need to devellop more about the obligatory pseudopregnancy in this species to explain why you did not see any differences between females during pregnancy period. Also it would be nice to explain if the females were at visual/olfactive distance from each others. Pheromones can definitively influence hormonal secretions.
- Line 313: What other event could have happen then? Meteorologic, high or low temperature, heavy rain...
Author Response
Reviewer #2
- add scientific name in the title
REPLY: Added scientific name to the title as requested.
- Always good to spell out number between 0 and 9. To change in the whole document
REPLY: Spelled out 0 to 9 where appropriate as requested.
- Line 16: change to "...with no apparent effect of reproduction on…"
REPLY: Changed to “…with no apparent effect of reproduction on…” as requested.
- Line 21: Change to "physiological stress"
REPLY: Changed to “physiological stress” as requested.
- Line 41: metabolic
REPLY: Changed to “metabolic” as requested.
- Line 66: "are complex"
REPLY: Changed to “are complex” as requested.
- Line 307: your study should not be considered as a reference
REPLY: Deleted our reference to “this study” as requested.
- Line 313: "six of the seven males…"
REPLY: changed to “six of the seven males” as requested.
- Line 22: Glucocorticoid metabolism is not the only parameter influenced by stress. It is necessary to add a term such as " ...are partly reflected..."
REPLY: Changed to “…are partly reflected…” as requested.
- Line 34: Corticoid alone is not a reliable parameter of stress therefore difficult to draw any conclusion about physiological stress
REPLY: Changed to “did not appear to statistically influence measures of fGCMs in captive coyotes” as suggested.
- Line 69: Gc are not alone good indicators of stress, hence explaining the sometimes contradictory results in different studies
REPLY: Added the following sentence “Additionally, measures of glucocorticoid metabolites may not in of themselves always be good indicators of stress, hence explaining the contradictory results found from different studies” to address this comment.
- 2.1: This part is supposed to be about the subjects but you are explaining about subject, housing, sample collection and your timeline decision. It should be split into different parts.
REPLY: Split these into two sections “Study Subjects and Sample Collection” and “Time Periods for Analysis” as requested.
- Line 117: add information about glitters (type, brand and so on) it will be interesting for the readers and for potential researcher willing to use the same method
REPLY: Added manufacturer of the glitter as requested; now reads “(Glitterex Corporation, Cranford, New Jersey)
- Line 127: add which temperature the sample where frozen at
REPLY: Added “…at -80 C” as requested.
- Line 140: it should be explained why the decision to remove the pups at 8-10 weeks of age instead of following natural dispersion between 6-9 months old for males and females staying with the parents longer.
REPLY: The animals are in a captive facility and therefore natural dispersion is not possible. We remove the pups for colony management and reducing competition/injuries from having too many animals in a confined space.
- 2.2: Need to be more detailed: preparation of the sample pre-extraction. Steps of extraction and ELISA... Also need to explain how the method was optimised for coyotes.
REPLY: I have added more details of fecal extractions, ELISAs, etc. as requested by both reviewers.
- Line 189 to 191: the choice of calling the time period "early pregnancy" is confusing when associated to non pregnant animal or males in the same sentence. It would be good to call the period with a specific code in the text like you did for the graphics.
REPLY: I could not find the term “early pregnancy” anywhere in the manuscript.
- Line 218: higher Cg during pregnancy is due to both mother and foetuses production this should be explained
REPLY: The GC level measured in the female was before embryos were released and before development of the fetus, and thus did not contribute to the higher CG observed.
- Discussion: there is no mention of the limitations of your study. It is essential that you talk about it for example small number of animals, only one sample collected per week creating gap into the data (ideally this kind of study should collect between two and three sample a week to decrease bias in variation). Study only over one breeding season... Gc not the ideal parameters for stress evaluation and mentioning about the importance of holistic approach via the use of the allostatic load instead.
REPLY: I have added more information to various parts of the discussion and conclusion demonstrating that this is the first and only study examining fGCM levels in coyotes covering an entire year; see highlights in yellow for added information and discussion. Granted more samples of coyotes and scats would have been great to have, but honestly, coyotes are not white lab rats and require large, expensive pens covering many acres.
-Line 266: you mention wild cheetah when the paper you reference to is about captive animals.
REPLY: Deleted this sentence as it does not refer to wild cheetahs. Thank you for catching my mistake.
- Line 292: need to develop more about the obligatory pseudopregnancy in this species to explain why you did not see any differences between females during pregnancy period. Also it would be nice to explain if the females were at visual/olfactive distance from each others. Pheromones can definitively influence hormonal secretions.
REPLY: Added more discussion (lines 294-297 and 300-304) to address this comment.
- Line 313: What other event could have happen then? Meteorologic, high or low temperature, heavy rain...
REPLY: After examining the activity logs for the facility, we found that these spikes were directly related to the capture and handling on the pups born to the five pairs on this study, plus the pups from three neighboring pens. This information has been added to the discussion (lines 322-328).
Round 2
Reviewer 2 Report
Comments and Suggestions for Authors
Dear authors,
Thanks a lot for taking into account my comments and making the modifications in your text. The paper looks good.
However one of my comment was not approached yet:
Line 245 to 249: the choice of calling the time period "early gestation" is confusing when associated to non pregnant animal or males in the same sentence. It would be good to call the period with a specific code in the text like you did for the graphics.
I mistakenly written pregnancy instead of gestation. However it does not make the sentence less confusing.
Author Response
Please see my response to Reviewer #2's comment below.
Line 245 to 249: the choice of calling the time period "early gestation" is confusing when associated to non pregnant animal or males in the same sentence. It would be good to call the period with a specific code in the text like you did for the graphics.
REPLY: I revised this sentence to now read "...occurred during 4-6 weeks of gestation....pups during the first 1-3 weeks of gestation..." (lines 245-248). And the next sentence was revised to now read "...occurred during the first 1-3 weeks of gestation....pups occurred during 4-6 weeks of gestation..." (lies 248-251) as requested to use the specific code in the text as was used in the graphics. These revisions are highlighted in BLUE in the revised manuscript. Thank you for pointing out this dissimilarity in terminology...